# Is the Pandemic Wearing Us Out? A Cross-Sectional Study of the Prevalence of Fatigue in Adult Twins without Previous SARS-CoV-2 Infection

**DOI:** 10.3390/jcm11237067

**Published:** 2022-11-29

**Authors:** Sophia Kristina Rupp, Katja Weimer, Miriam Goebel-Stengel, Paul Enck, Stephan Zipfel, Andreas Stengel

**Affiliations:** 1Department of Psychosomatic Medicine and Psychotherapy, University Hospital Tübingen, 72076 Tübingen, Germany; 2Department of Psychosomatic Medicine and Psychotherapy, Ulm University Medical Center, 89081 Ulm, Germany; 3Department of Internal Medicine, Helios Kliniken GmbH, 78628 Rottweil, Germany; 4Charité Center for Internal Medicine and Dermatology, Department for Psychosomatic Medicine, Charité—Universitätsmedizin Berlin, Corporate Member of Freie Universität Berlin, Humboldt-Universität zu Berlin and Berlin Institute of Health, 10117 Berlin, Germany

**Keywords:** brain fog, mental health, psychosomatic, somatoform, twins

## Abstract

During the pandemic, mental health was not only impaired in people after a SARS-CoV-2 infection, but also in people without previous infection. This is the first study on twins without prior SARS-CoV-2 infection to estimate the influence of genetic components and shared as well as individual environments on pandemic-associated fatigue. The study sample included 55 monozygotic and 45 dizygotic twin pairs. A total of 34.5% reported an increase in fatigue since the pandemic. A significant correlation was shown between the responses within monozygotic (χ^2^[1] = 11.14, *p* = 0.001) and dizygotic pairs (χ^2^[1] = 18.72, *p* < 0.001). In all pandemic-associated fatigue dimensions, individual environment (ranging from e^2^ = 0.64 to e^2^ = 0.84) and heritability (ranging from h^2^ = 0.32 to h^2^ = 1.04) seem to have the highest impact. The number of comorbidities significantly correlated with physical fatigue (Spearman’s *ρ* = 0.232, *p* < 0.001) and psychological impairment due to pandemic measures with the total fatigue score (Spearman’s *ρ* = 0.243, *p* < 0.001). However, calculated ANCOVAs with these significant correlations as covariates showed no significant influence on the mean values of the respective fatigue dimensions. Susceptibility to pandemic-associated fatigue may be genetically and environmentally determined, while intensity is also influenced by individual components. The prevalence of fatigue is high even in individuals without prior SARS-CoV-2 infection. Future mental health prevention and intervention programs should be implemented to alleviate the impact of the pandemic on the global population.

## 1. Introduction

During the pandemic, it has become increasingly clear that mental health is not only impaired in people after a SARS-CoV-2 infection, but also in people without prior infection [1,2]. Here, symptoms such as fatigue, sleep disturbances, mood swings or loss of interest predominate [3] and are as frequent as in infected persons [4,5]. These findings indicate that prevention, promotion and intervention programs for mental health problems should be conducted to alleviate the impact of the COVID-19 pandemic on the global population.

Twin studies are an essential methodological tool for the assessment of genetic and environmental influences. Monozygotic (MZ) twins that have grown up together have a comparable environmental influence and share almost 100% of their genes. Dizygotic (DZ) twins that have grown up together are also exposed to a comparable influence of environmental factors but share only about 50% of their genetic make-up [6]. Therefore, comparing traits between MZ and DZ twins can help to identify the crucial part of genetics and environmental factors. The study of concordance and discordance in MZ compared to DZ twins is one methodological way to determine the contribution of genes to disease development and progression.

Some twin studies have already examined mental health during the pandemic and shown that depression [7,8], anxiety [7,8,9] as well as perceived stress [8,9] increased during the COVID-19 pandemic, while optimism and a sense of life meaning declined [10]. These changes were explained, on one hand, by the hypothesis of genetic stability between waves and environmental discontinuity due to changes in living conditions during the pandemic [7], and, on the other hand, by the theory that subjects’ genetic constitution behaves rather dynamically and becomes more apparent over time, e.g., due to social isolation [11]. However, reports are inconsistent, as there are also studies implying that the lockdown and the pandemic had little to no impact on the mental health of participating twins [12,13].

Regarding fatigue, the most common persistent symptom after COVID-19, no studies have yet used twins without prior SARS-CoV-2 infection to investigate or record the increase in fatigue or its severity since the pandemic. Thus, we enrolled adult MZ and DZ twins without prior SARS-CoV-2 infection to assess the prevalence and changes in fatigue during the pandemic and investigate whether those can be explained by genetic, shared environmental and/or individual components.

## 2. Materials and Methods

### 2.1. Participants and Procedure

The study sample is derived from the German TwinHealth Twin Registry at the University Hospital of Tübingen [14]. This Twin Registry currently contains information on more than 400 adult twin pairs of different ages and geographical areas who have given their written consent to be contacted to participate in TwinHealth-related research projects.

Inclusion criteria were fluency in German and participation of both twins in the online survey. The sample includes 155 twin pairs who participated in a COVID-19 online survey in early 2022. The online survey included questions on the following topics, among others: SARS-CoV-2 infection in the past, several fatigue dimensions, and changes in fatigue since the pandemic. Of these 155 twin pairs, 100 pairs reported that none of the twins had previously contracted SARS-CoV-2.

This study was approved by the Ethical Review Board of the University Hospital of Tübingen (project No. 174/2020BO1) and was conducted in accordance with the Declaration of Helsinki.

### 2.2. Zygosity Assessment

To assess zygosity, we used questions on similarity of appearance between twins, confusion by strangers and previous genetic zygosity tests, as already described in a previous study [15]. This method has been shown to reliably discriminate between MZ and DZ twins [16,17]. A zygosity score between 0 (high dissimilarity) and 20 (high similarity) was calculated [15], and a score of ≥10 or higher was assumed to be MZ, while a score of <10 was indicative of DZ. The scores were compared with the self-report on the zygosity of the twins and agreed upon in all cases.

### 2.3. Measures

A self-designed questionnaire was used, which included questions on socio-demographic characteristics. In addition, the participants were asked about a previous or current SARS-CoV-2 infection and the current vaccination status. Moreover, social withdrawal due to fear of SARS-CoV-2 infection and psychological impairment due to pandemic measures were rated from “0 = not at all” to “10 = very strongly”. Lastly, participants were asked to answer with “Yes” or “No” to the question “Do you feel that fatigue, exhaustion, and reduced resilience have increased in you since the COVID-19 pandemic?”.

Information on comorbidities of the twins was extracted from the baseline data set of the TwinHealth Twin Registry of the University Hospital of Tübingen.

To assess the severity and different domains of the fatigue syndrome, the Multidimensional Fatigue Inventory (MFI) was used. This instrument consists of 20 items on 5 dimensions of fatigue syndrome: general fatigue, physical fatigue, mental fatigue, reduced activity and reduced motivation [18]. For each of the 20 items, the respondent is offered 5 response options ranging from “Yes, that is true” to “No, that is not true”. The internal consistency of the instrument was satisfactory; Cronbach’s α ranged from 0.76 to 0.88 for the different fatigue dimensions. The total score of the MFI reached an α coefficient of 0.95.

### 2.4. Statistical Analyses

For descriptive analyses of the collected data, statistical measures such as mean with associated standard deviation as well as the range were provided for metric variables. For categorical variables, absolute and relative frequencies were calculated. The Kolmogorov–Smirnov test and visual inspection of quantile–quantile plots were used to assess the normal distribution of the variables. Internal consistency was assessed by using Cronbach’s α. Differences between groups were analyzed with Student’s *t*-tests. Twin data were arranged according to the registration order at the TwinHealth Registry, i.e., the twin registered first was assigned the suffix A, while the other twin was assigned B.

Independence of categorical variables with an expected cell frequency of >5 was determined using the χ^2^-test. 

The extent to which changes in fatigue since the pandemic affected the level of mean fatigue scores was assessed using one-way analyses of covariance (ANCOVA) with sex, age, psychological impairment from the pandemic and number of comorbidities as covariates. 

Intra-class correlations (ICCs, one-way random, single measurement) were calculated separately for MZ and DZ twins to more precisely quantify and differentiate percentage shares of genetic influences, as well as common environment and personal experiences on the examined traits [19,20]. ICCs are a common method for the estimation of inter-rater reliability in the assessment of an outcome or trait [19]. ICCs can take values between −1 and 1 (−1 < ICC < +1), but generally range from 0 to 1 and were interpreted according to Cicchetti as follows: <0.40 = poor, between 0.40 and 0.74 = moderate to good, between 0.75 and 1.00 = excellent [21]. If the variance within pairs is greater than the variance between pairs, the ICCs take on negative values. Accordingly, negative ICCs should be interpreted as no correlation [22] and are taken as zero in subsequent calculations using the Falconer’s formula [19,20]. For determining the percentages of genetic and environmental influence, the Falconer’s formula was used [23,24]. The theoretical assumptions of this model are as follows: (1) MZ twins share 100% of their genetic make-up; (2) DZ twins share 50% of their genes; (3) MZ and DZ twins that have grown up together share 100% of their common environment; (4) Other effects such as non-shared environment, individual learning experiences and measurement errors contribute to differences within twin pairs. Using the Falconer’s formula, heritability (h^2^ = 2 * [rMZ − rDZ]), shared environmental effects (c^2^ = 2 * rDZ − rMZ) and non-shared or individual environmental effects (e^2^ = 1 − rMZ) were estimated based on the calculated twin correlation. The relative influences of heritability and shared and individual environment consequently add up to 100%. High correlations within MZ twins, which are at the same time higher than correlations within DZ twins, suggest the presence of a genetic effect. High correlations in both MZ and DZ twins indicate a major role of common environmental influences, while low correlations indicate that non-common or individual environmental influences are responsible for the twins’ dissimilarity. 

All statistical analyses were performed with IBM SPSS Statistics for Windows, Version 27.0 (IBM Corp., Armonk, NY, USA). Significance level was set at *p* < 0.05 for all analyses.

## 3. Results

### 3.1. Study Population

The study sample included 200 twins (55 MZ and 45 DZ pairs). Of the 55 MZ pairs, 43 pairs were female, and 12 pairs were male, while 45 DZ pairs included 22 female, 5 male, and 18 opposite-sex pairs. The MZ twins were 45.95 ± 16.61 years old and the DZ twins were 46.96 ± 17.20 years old. Comorbidities of the study sample were classified by organ system (Table 1).

### 3.2. Change in Fatigue since the COVID-19 Pandemic and Influencing Factors

Of all twins participating in this trial, 131 (65.5%) answered “No” and 69 (34.5%) answered “Yes” to the question of whether fatigue, exhaustion or reduced resilience had increased since the COVID-19 pandemic. Of the group that answered “Yes”, 54 (78.3%) were females and 15 (21.7%) males. Of the group that answered “No”, 94 (71.8%) were females and 37 (28.2%) males.

A χ^2^-test was performed to compare the responses of twin A and twin B. The results show a strong significant correlation between the responses within MZ pairs; χ^2^(1) = 11.14, *p* = 0.001, *φ* = 0.45. A total of 13 pairs concordantly answered “Yes”, and 28 pairs concordantly answered “No”. A total of 14 pairs gave discordant answers. Within the responses from the DZ pairs, there was also a strong significant correlation; χ^2^(1) = 18.72, *p* < 0.001, φ = 0.65. A total of 11 pairs answered “Yes”, and 27 pairs answered “No” in concordance. A total of 7 pairs gave discordant answers.

There was no statistically significant difference in the mean scores of fatigue symptoms between the group that answered “Yes” and the group that answered “No” (Table 2).

No significant association was found between increase in fatigue since the pandemic and sex (*r* = −0.07; *p* = 0.32). Female sex was coded 1, and male sex was coded 2. We found significant positive correlations between sex and several fatigue dimensions (Table 3).

Regarding the age of the participants, no significant correlation was found with increase in fatigue since the pandemic (*r* = −0.02; *p* = 0.79). Age was statistically significantly correlated with physical fatigue and reduced motivation.

In terms of vaccination status, of 200 twins, 8 (4%) had not been vaccinated against SARS-CoV-2, 0 (0%) had been vaccinated one time, 4 (2%) had already received two vaccinations and 188 (96%) twins had been vaccinated three times. Vaccination status within the twin pairs showed a significant correlation with each other (*ρ* = 0.489; *p* < 0.001). No significant correlation was found between an increase in fatigue since the pandemic and the number of vaccinations (*r* = 0.08; *p* = 0.25). Regarding the different dimensions of fatigue, we also found no correlation with the number of vaccinations.

The number of comorbidities of the 200 twins ranged from 0 to 5. Mean was 1.28 ± 1.19 comorbidities. No significant correlation was found between an increase in fatigue since the pandemic and the number of comorbidities (*r* = 0.14; *p* = 0.05). We found significant positive correlations between the number of comorbidities and several fatigue dimensions (Table 3).

On average, 200 twins reported a mean score of 5.77 ± 2.99 in social withdrawal (range 0–10) due to a fear of SARS-CoV-2 infection, while psychological impairment due to pandemic measures had a mean score of 3.90 ± 2.71 (range 0–10). Twin A’s and twin B’s fear of infection showed a significant correlation with each other (*ρ* = 0.248; *p* = 0.013), as well as the psychological impairment (*ρ* = 0.385; *p* < 0.001). There was no significant correlation for the increase in fatigue since the pandemic and social withdrawal due to fear of SARS-CoV-2 infection (*r* = −0.087; *p* = 0.22), as well as the psychological impairment due to pandemic measures (*r* = 0.09; *p* = 0.19). We found no significant correlations between a fear of infection with SARS-CoV-2 and the different fatigue dimensions. In terms of psychological impairment due to pandemic measures, there were significant positive correlations with several fatigue dimensions (Table 3). In particular, the total fatigue score was statistically significantly positively correlated (*ρ* = 0.243, *p* ≤ 0.001) with psychological impairment due to pandemic measures (Figure 1).

To further examine whether the mean values of different fatigue dimensions of the group that reported an increase in fatigue since the pandemic and the group that reported no change were confounded by possible influencing factors, we integrated significant bicorrelations (according to Table 3) as covariates in one-way ANCOVAs (Table 4).

After adjusting for sex, age, psychological impairment due to pandemic measures and number of comorbidities, no statistically significant difference in mean values on any fatigue dimension was found for the two groups.

### 3.3. Concordance of Fatigue Symptoms during COVID-19 Pandemic within Pairs

Of the total 24 pairs who concordantly reported an increase in fatigue and exhaustion since the pandemic, in the MZ pairs (*n* = 13) the means of the different fatigue dimensions ranged from 10.16 to 11.35. The mean of total fatigue score was 53.54 ± 18.26 (Table 5). For the DZ pairs (*n* = 11), the mean ranged from 10.68 to 12.55 for the fatigue dimensions. The mean of total fatigue score was 58.50 ± 15.75 (Table 5). 

Reduced activity significantly correlated within the MZ pairs (intra-class correlation coefficient, ICC(1) = 0.52; *p* = 0.02) but not between the DZ pairs (ICC(1) = −0.14; *p* = 0.67). There were no statistically significant correlations of other fatigue dimensions within MZ or DZ pairs (Table 5).

### 3.4. Genetic, Common and Individual Environment Contributions to Fatigue Dimensions

Among the twin pairs who concordantly reported an increase in fatigue and exhaustion since the pandemic, the individual environment and heritability appeared to have the strongest influence on all fatigue dimensions, while there seems to be little to no influence of common environmental effects (Figure 2).

A non-shared or individual environment had the highest value of 0.84 for reduced motivation. For general fatigue, physical fatigue, mental fatigue and overall fatigue the scores ranged from 0.64 to 0.80. 

In contrast, high heritability was estimated for reduced activity (h^2^ = 1.04) and for general fatigue (h^2^ = 0.72).

## 4. Discussion

This is the first study of previously SARS-CoV-2 uninfected MZ and DZ twins recording changes in fatigue and estimating the variances of different fatigue dimensions during the COVID-19 pandemic explained by genetic, shared environmental and individual components. Data were collected by having both twins participate in the survey at one time point. Regarding changes in fatigue, exhaustion or reduced resilience since the COVID-19 pandemic, almost two-thirds reported no changes, while about one-third reported an increase since the pandemic, which is in line with other studies reporting an equally frequent prevalence of fatigue in people with prior SARS-CoV-2 infection [3,4,5]—raising the question of whether fatigue is truly a symptom of COVID-19 or whether the pandemic is exhausting us all and previous results are biased because infected subjects are being studied more closely. By comparing the responses of twin A and twin B, we found a significant concordance within both MZ and DZ pairs, which may indicate an influence of genetic effects and shared environment on susceptibility of pandemic-related fatigue. Similarly, heritability has been estimated as an influencing factor for fatigue in general in other twin studies [25,26,27,28].

Regarding the severity of the different pandemic-related fatigue dimensions within twin pairs, we found a moderate-to-good ICC for reduced activity in the MZ pairs, suggesting that genetic effects may play a role here. In contrast, the ICCs within the DZ pairs only showed negative values in all fatigue dimensions, which we interpreted as no correlation according to Cicchetti [20,21]. However, the ICCs in the MZ pairs were also rather low in the other fatigue dimensions. These findings suggest that the influence of individual or non-shared environment on the different fatigue dimensions is high, and that genetics or shared environment may play a lesser role. The estimates of heritability (h^2^), common (c^2^) and individual (e^2^) environmental influences confirm our findings: of the twin pairs who concordantly reported an increase in fatigue and exhaustion since the pandemic, the individual environment appears to have the strongest influence in most fatigue dimensions. In contrast, for general fatigue and reduced activity, genetic effects seem to play a role. Overall, we found little to no influence of common environmental effects for all fatigue dimensions, which is inconsistent with a twin study of 2010 which found that leaving school early, poor living standards, negative life events and poor parental care mediated fatigue through shared environmental influences [29]. Since there are no previous studies of pandemic-associated fatigue within twin pairs without prior SARS-CoV-2 infection, we further compared our results with other twin studies of fatigue associated with COVID-19, depression or anxiety. Here, we found that our results are consistent with these twin studies, as they also reported a moderately heritable component underlying fatigue [26,28,30,31]. However, other twin studies estimate a lower influence of the non-shared environment than our results show [25,26,27,28,30]. Nevertheless, the calculations of h^2^, c^2^ and e^2^ should be interpreted as estimates indicating the direction of the effects, but not as absolute values. Further research is needed to better understand the genetic component and identify potential candidate genes contributing to fatigue.

When comparing the group experiencing an increase in fatigue since the COVID-19 pandemic and the group reporting no change, there were no significant differences in severity in any of the fatigue dimensions. Only slightly higher mean values were recorded in some dimensions of the group that reported pandemic-associated fatigue. Overall, both groups achieved rather average values in the scores of the various fatigue subscales; compared to the general German population, the scores were slightly elevated [32], compared to patients after the COVID-19 pandemic, for whom the scores were lower [33]. Possible explanations for the similar fatigue scores in both groups could be, for example, the subjective perception of increased fatigue since the pandemic caused by less leisure activities due to, e.g., pandemic measures and no fatigue as per se. On the other hand, the MFI may not be the optimal instrument to measure pandemic-associated fatigue. Overall, total scores rather than subscale scores seem to be more appropriate for assessing fatigue in the clinical setting [34]. An important factor to consider is that we used only one question that asked about change in fatigue since the pandemic. This information is based on self-reports by participants, and to date there is no information about the validity or reliability of this single question. Since this study was cross-sectional, fatigue was only assessed at one point in time and there was no measurement of fatigue pre-pandemically. Thus, conclusions about the actual increase in fatigue, exhaustion or reduced resilience should be drawn with caution. Furthermore, when interpreting the results of our study, it should be considered that our study population probably consists mainly of people suffering from sub-syndromal fatigue and not primarily from fatigue syndrome. This could also be a reason why our results show such a strong influence of the individual environment. 

There are already some studies that have examined changes in mental health during the pandemic in twins without prior SARS-CoV-2 infection, more precisely changes in depression [7,8], anxiety [7,8,9] and stress [8,9]. Researchers have explained these changes on the one hand by the hypothesis of environmental discontinuity due to changed living conditions during the pandemic [7], and, on the other hand, by the theory that genetic constitution tends to behave dynamically and became more apparent over time, e.g., through social isolation [11]. Since in our study population the individual environment seems to play a major role in the intensity of pandemic-associated fatigue, we took a closer look at this group to identify factors that may contribute to change in fatigue associated with the COVID-19 pandemic.

It is noticeable that the proportion of women, at 78.3%, is significantly higher than the proportion of men in this group. Several studies have identified female gender as a potential predisposing factor for fatigue [35,36,37,38], which may explain the gender ratio in this group. Other risk factors reported include older age [37], comorbidities [37,38,39], depression and anxiety [28,38,39,40,41], educational attainment [39] and socioeconomic status [39,42]. Overall, the pathogenesis of fatigue may be best explained by a biopsychosocial model [42], with genetic factors [37] and previous infectious or autoimmune diseases [43] also being influencing factors that may contribute to the development of fatigue or account for increased susceptibility. However, current evidence on precipitants or triggers is still inconsistent [39,44,45], indicating potential areas of research that require further exploration to base future practice on the best scientific evidence.

Regarding individual environmental factors, several studies reported fatigue as one of the most common temporary effects following all types of SARS-CoV-2 vaccines [46,47,48]. Other researchers report that the mRNA-based vaccines are more likely associated with local adverse effects, while viral vector-based vaccines are more likely to cause systemic side effects such as fatigue [48]. Patients also reported persistent fatigue at 7 days [49,50,51], 14 days [52] and 3 months [53] after vaccination. Our results are inconsistent with these reports: We found no significant correlation between the occurrence of pandemic-associated fatigue and vaccination status nor between the severity of the different fatigue dimensions and vaccination status. However, we did not consider the exact time of the twins’ last vaccination, nor the type of vaccine in our calculations. Nevertheless, in our study, vaccination status was significantly concordant within the pairs, which may explain why the change in fatigue since the pandemic was often similar within the pairs. 

Furthermore, in line with previous studies [37,39], we show that participants who suffer from pandemic-related fatigue tend to have a higher number of comorbidities. Additionally, our results show that with an increasing number of comorbidities, the severity of fatigue also increases. 

Moreover, our results show that subjects who feel psychologically impaired due to pandemic measures show significantly higher scores on certain fatigue dimensions. We also investigated whether social withdrawal due to fear of SARS-CoV-2 infection or the severity of psychological impairment due to pandemic measures influences the occurrence of pandemic-associated fatigue: Here, we found no correlation, although several studies have shown that fatigue [54,55] and mental health problems [2,56,57] during the pandemic increase with fear of SARS-CoV-2 infection. In general, the occurrence and severity of fatigue can be favored by the presence of certain personality traits, e.g., neuroticism [58,59,60].

To further examine whether the mean values of the different fatigue dimensions of the group that reported an increase in fatigue since the pandemic and the group that reported no change were confounded by these potentially influencing factors, we included these variables as covariates in one-way ANCOVAs. After adjusting for these potential influencing factors, no statistically significant difference was found in the mean scores of all fatigue dimensions for either group. These results may challenge the MFI or the single question on the increase as appropriate tools.

Finally, some limitations of this study should be mentioned: the exclusive use of self-reports in absence of direct contact with the twins, and the only one-time data collection. In addition, we did not determine zygosity by genetic testing, but relied on the twins’ self-reports of genetic testing and questions about the similarity and dissimilarity of the twins and compared the results with the twins’ self-reports of zygosity. Although this procedure showed high agreement with genetic tests [16,17], it is not as precise as genetic tests. Furthermore, it should be kept in mind that we only used a single question to assess the change in fatigue since the pandemic. This question has not yet been validated to measure a change in fatigue since the pandemic, warranting a cautious interpretation of the data. In addition—to increase the acceptance of the brief survey—we did not assess educational status, socioeconomic status and time since vaccination, although all these factors can have an influence on the severity and occurrence of fatigue [37,39,42,48]. The strengths of our study are the following: First, our study sample included only twin pairs without prior SARS-CoV-2 infection, which allowed for an assessment of genetic and environmental influences on pandemic-associated fatigue. This concordance also allowed us to draw conclusions about possible risk and influencing factors underlying fatigue. Furthermore, our study population had a moderate size and the distribution of MZ and DZ twins was almost equal.

In summary, our study results suggest that susceptibility to pandemic-associated fatigue may be genetically and environmentally determined, which explains why the changes in fatigue since the pandemic were significantly similar within twin pairs, whereas intensity and severity of the different fatigue dimensions are also influenced by individual environmental factors such as number of comorbidities. We were also able to show that with increasing psychological impairment due to pandemic measures, the severity of fatigue also increases. All in all, pandemic-associated fatigue without prior SARS-CoV-2 infection is a common complaint and requires further investigation to understand the exact pathogenesis and to successfully prevent or treat it. As we demonstrated, mental health problems increased during the pandemic even in individuals without prior SARS-CoV-2 infection. Therefore, prevention and intervention programs for mental disorders and avoidance of complete lockdowns should be implemented in the future to mitigate the further impact of the COVID-19 pandemic on the population.

## Figures and Tables

**Figure 1 jcm-11-07067-f001:**
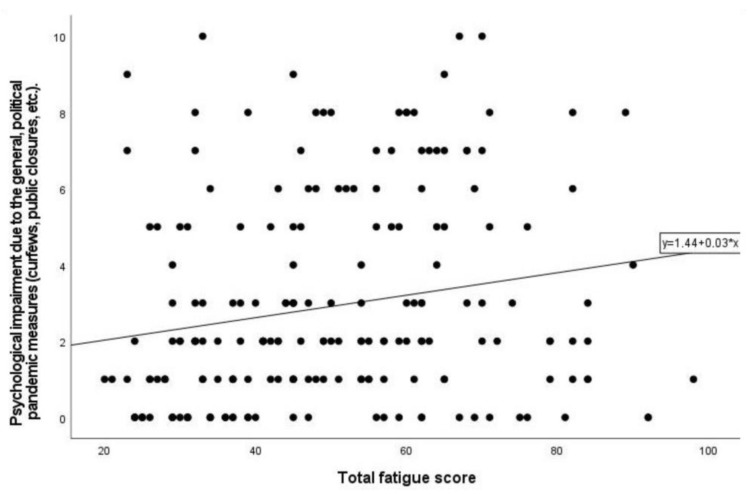
Correlation between psychological impairment due to general, political pandemic measures (curfews, public closures, etc.) and total fatigue score.

**Figure 2 jcm-11-07067-f002:**
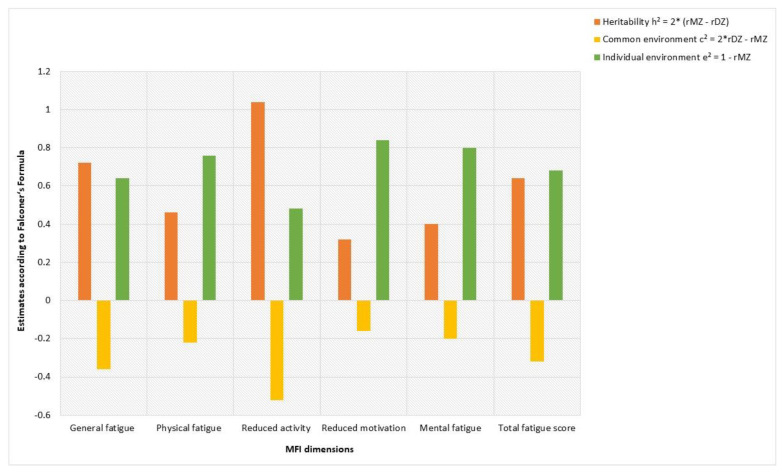
Estimates of heritability (h^2^), common (c^2^) and individual environmental (e^2^) effects on fatigue dimensions of twin pairs (*n* = 24) concordantly experiencing increased fatigue since the COVID-19 pandemic according to the Falconer’s formula. Notes: MFI = Multidimensional Fatigue Inventory.

**Table 1 jcm-11-07067-t001:** Demographics and baseline characteristics of the study population.

**Gender**		
	*n*	%
**Female**	148	74
**Male**	52	26
**Zygosity**		
	*n*	%
**Monozygotic**	110	55
**Dizygotic**	90	45
**Age (years)**		
	Monozygotic	Dizygotic
**Mean**	45.95	46.96
**SD**	16.61	17.20
**Minimum**	19	19
**Maximum**	77	82
**Comorbidities**		
	*n*	%
**Thyroid gland**	25	12.5
**Lungs**	26	13.0
**Cardiovascular system**	36	18.0
**Kidney**	12	6.0
**Pancreas**	4	2.0
**Gastrointestinal tract**	49	24.5
**Spine**	45	22.5
**Mental**	26	13.0

**Table 2 jcm-11-07067-t002:** Differences between fatigue scores of the group experiencing an increase in fatigue since the pandemic and the group that reported no difference in context of the pandemic. (*n* = 200).

	General Fatigue		Physical Fatigue		Reduced Activity		Reduced Motivation		Mental Fatigue		Total Fatigue Score	
Increase in fatigue, exhaustion or reduced resilience since the pandemic	Yes (*n* = 69; 34.5%)	No (*n* = 131; 65.5%)	Yes (*n* = 69; 34.5%)	No (*n* = 131; 65.5%)	Yes (*n* = 69; 34.5%)	No (*n* = 131; 65.5%)	Yes (*n* = 69; 34.5%)	No (*n* = 131; 65.5%)	Yes (*n* = 69; 34.5%)	No (*n* = 131; 65.5%)	Yes (*n* = 69; 34.5%)	No (*n* = 131; 65.5%)
Mean	10.91	10.42	9.72	9.84	9.54	10.02	9.78	9.21	10.46	9.73	50.42	49.21
SD	3.834	4.004	3.933	4.383	3.909	3.963	3.827	3.920	4.020	4.384	16.953	18.168
t(df) = T	t(198) = −0.840		t(198) = 0.183		t(198) = 0.816		t(198) = −0.997		t(198) = −1.165		t(198) = −0.460	
*p* (2-sided)	0.402		0.855		0.415		0.320		0.245		0.646	

Student’s *t*-test (for independent samples).

**Table 3 jcm-11-07067-t003:** Correlations between different fatigue dimensions and sex, age, number of vaccinations against SARS-CoV-2, fear of infection as well as mental impairment due to pandemic measures. Reported as Spearman’s correlation coefficients *ρ*. (*n*= 200).

Item	Sex	Age	Number of Vaccinations	Fear of Infection with COVID-19	Psychological Impairment due to Pandemic Measures	Number of Comorbidities
General fatigue	0.094	-0.029	0.036	0.066	0.229 ***	0.191 **
Physical fatigue	0.139 *	0.160 *	0.038	0.118	0.135	0.232 ***
Reduced activity	0.156 *	-0.025	0.002	0.080	0.207 **	0.137
Reduced motivation	0.191 **	0.142 *	0.080	0.077	0.231 ***	0.158 *
Mental fatigue	0.110	−0.035	0.069	0.065	0.232 ***	0.134
Total fatigue score	0.157 *	0.044	0.055	0.091	0.243 ***	0.199 **

* *p* < 0.05. ** *p* < 0.01. *** *p* ≤ 0.001.

**Table 4 jcm-11-07067-t004:** Unadjusted and adjusted mean values of different fatigue dimensions calculated as one-way ANCOVAs. (*n* = 200).

		Unadjusted		Adjusted		
Increase in Fatigue Since the Pandemic	*n*	M	SD	M	SE	*p*
General fatigue						
No	131	10.42	4.00	10.55	0.33	0.82
Yes	69	10.91	3.83	10.67	0.46	
Physical fatigue						
No	131	9.84	4.38	9.89	0.36	0.65
Yes	69	9.72	3.93	9.61	0.50	
Reduced activity						
No	131	10.02	3.96	10.03	0.34	0.37
Yes	69	9.54	3.91	9.51	0.47	
Reduced motivation						
No	131	9.21	3.92	9.24	0.33	0.40
Yes	69	9.78	3.83	9.72	0.45	
Mental fatigue						
No	131	9.73	4.38	9.78	0.37	0.36
Yes	69	10.46	4.02	10.35	0.51	
Total fatigue score						
No	131	49.21	18.17	49.60	1.49	0.97
Yes	69	50.42	16.95	49.68	2.07	

**Table 5 jcm-11-07067-t005:** Fatigue symptoms in MZ and DZ twin pairs concordantly experiencing increased fatigue since the COVID-19 pandemic (reported as mean ± standard deviation) and intra-class correlations (reported as ICC coefficients and 95% CI).

Item	MZ Pairs (*n* = 13)			DZ Pairs (*n* = 11)		
	Twin A	Twin B	ICC (95%CI)	Twin A	Twin B	ICC (95%CI)
General fatigue	11.38 ± 4.56	11.31 ± 3.99	0.36 [(−0.20–0.74)	13.55 ± 3.67	11.55 ± 3.11	−0.28 (−0.73–0.34)
Physical fatigue	9.69 ± 4.42	10.62 ± 4.41	0.24 (−0.32–0.68)	11.09 ± 4.16	11.45 ± 3.36	0.01 (−0.55–0.58)
Reduced activity	10.77 ± 4.59	9.69 ± 3.90	0.52 * (0.01–0.82)	11.91 ± 3.33	11.18 ± 4.02	−0.14 (−0.65–0.47)
Reduced motivation	11.00 ± 3.46	10.00 ± 4.47	0.16 (−0.39–0.64)	10.27 ± 3.23	11.09 ± 5.11	−0.01 (−0.56–0.57)
Mental fatigue	12.15 ± 3.76	10.46 ± 2.90	0.20 (−0.36–0.66)	13.18 ± 3.79	11.73 ± 3.95	−0.20 (−0.68–0.42)
Total fatigue score	55.00 ± 18.70	52.08 ± 17.81	0.32 (−0.24–0.72)	60.00 ± 14.77	57.00 ± 16.72	−0.08 (−0.61–0.51)

Notes: * *p* < 0.05. DZ = Dizygotic twins. ICC = Intra-class correlation coefficient. MZ = Monozygotic twins.

## Data Availability

The raw data supporting the conclusions of this article will be made available by the authors, without undue reservation.

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
