# Peer review of "Is the Pandemic Wearing Us Out? A Cross-Sectional Study of the Prevalence of Fatigue in Adult Twins without Previous SARS-CoV-2 Infection"

_jcm, 2022, doi:10.3390/jcm11237067_

Round 1

Reviewer 1 Report

JCM-2050400

This is an interesting study to describe the prevalence of fatigue in adult twins without previous SARS- 3

CoV-2 infection. The manuscript could be improved if the following concerns could be addressed.

1.       Only one single question was used to measure the increase in fatigue since the pandemic. There were no measures or questions about fatigue before the pandemic. There is no reliability and validity information about the question. Based on the single question only, it is hard to know if fatigue is really increased or not since the pandemic.

2.       As shown in Table 2, the mean NFI total and subscale scores did not show significant differences between individuals with increase in fatigue or not. This may indicate the single question to measure the increase in fatigue is not valid or the association is confounded by age, sex, comorbidities, and other factors like depression, sleep problems. Adjusted analyses with age, sex, and commodities as covariates should be conducted as the 3 variables are available.

3.       Fatigue is multifactorial. Including demographics, physical and mental health problems. However, the study only reported its bicorrelations with number of vaccinations, fear of infection, psychological impairment due to pandemic measures, and the number of comorbidities.  

a.       This limitation should be well discussed.

b.       What are the comorbidities? They should be presented in detail as different comorbidities may have very different impacts on wellbeing.

c.       As age and sex are available, their associations with fatigue should be reported.

d.       Age and sex and/or comorbidities should be adjusted for in the statistical analyses as appropriate, such as findings presented in Table and Table 3.

4.       Table 1 is not necessary unless detailed comorbidities are added.

5.       Appendix A and B are not needed. Appendix A has been described in the method and Appendix B is a standardized scale.

6.       Some terms need to be corrected. For example, “this trial” (line#79) should be changed to be the survey or study. “Mental health prevention” (line#39, line#379) should be prevention of mental health problems or mental disorders.    

Reviewer 2 Report

This is an interesting paper, and the findings are novel.

Some minor changes could imprive the manuscript:

1. Table 2: Increase in fatigue, exhaustion, or reduced re-
silience since the pandemic- Please add percentages additionally to N values.

2. Table 3. The most study participants were vaccinated. What do these small values for number of vaccinations mean?

3. Resuts of correlations are not always correctly interpreted. " Twin A’s and twin B’s fear of infection showed a strong significant correlation with each other (ρ = 0.248; p =
0.013)". 
ρ = 0.248 is not a string correlation.  Values of r, 0-0.19 is regarded as very weak, 0.2-0.39 as weak, 0.40-0.59 as moderate, 0.6-0.79 as strong and 0.8-1 as very strong correlation.  This should be checked on other places where correlation results are interpreted.

Round 2

Reviewer 1 Report

My comments have been addressed.

Reviewer 2 Report

Thank you for your responses